# A Study of Robustness of Neural Nets Using Approximate Feature Collisions

## Abstract

In recent years, various studies have focused on the robustness of neural nets. While it is known that neural nets are not robust to examples with adversarially chosen perturbations as a result of linear operations on the input data, we show in this paper there could be a convex polytope within which all examples are misclassified by neural nets due to the properties of ReLU activation functions. We propose a way to find such polytopes empirically and demonstrate that such polytopes exist in practice. Furthermore, we show that such polytopes exist even after constraining the examples to be a composition of image patches, resulting in perceptibly different examples at different locations in the polytope that are all misclassified.

## 1 Introduction

Deep learning has achieved resounding success in recent years and is quickly becoming an integral component of many real-world systems. As a result of its success, increasing attention has focused on studying the robustness of neural nets, in particular to inputs that are deliberately chosen to yield an unexpected prediction. Various studies have shown the neural nets and other models are not robust to *adversarial examples* (Dalvi et al., 2004; Biggio et al., 2013; Szegedy et al., 2013), that is, examples from one class with minute perturbations that are designed to cause them to be classified as a different class. To make neural nets more resistant to these examples, it has been suggested (Goodfellow et al., 2014) to train the neural net specifically on these examples, which significantly reduces the error rate on this kind of examples.

One explanation for these adversarial examples (Goodfellow et al., 2014) is that linearity in the model makes it possible to construct adversarial perturbations that result in a large change in the dot product between the input and the parameters when the dimensionality of the input is high, even though the perturbation to each element of the input is small in magnitude. In this paper, we show the existence of another type of examples that would, too, yield unexpected predictions. Unlike adversarial examples, these examples arise from the properties of *non-linear* activation functions in neural nets. We show that there could be examples that look nothing like an example from the target class, but are classified confidently as that class. In fact, there could be infinitely many such examples, obtained by arbitrary convex combinations of different elementary examples. The key observation is that we could change any component in the pre-activation of a layer before ReLUs that has a negative value to any other negative value without changing the post-activation, and consequently keep all pre-activations and post-activations in later layers and therefore the predicted class label the same. We show that in general, for any neural net with a layer with ReLU activations, there is a convex (but possibly unbounded) polytope in the space of post-activations of the previous layer such that all input examples whose post-activation vectors fall in the interior or the boundary of the polytope will have identical activations in all later layers. Moreover, we develop a way of finding such a relaxed version of this polytope in practice, where the activations in later layers for examples within this polytope are only approximately equal. Finally, we show empirically that such a polytope exists near practically any input example that we tried, where all examples within this polytope result in the same (incorrect) classification. Surprisingly, such a polytope exists even if we were to constrain the input to be a composition of image patches, thereby ensuring that only macro-level changes to the input are permitted. We call examples in this polytope *approximate fea-*

*ture collisions* [1], to borrow terminology from the hashing literature, since these examples are quite different from the target example but share very similar activations in later layers.

## 2 METHOD

Consider any layer in a neural net with ReLU activations. Let $\mathbf{W} \in \mathbb{R}^{N \times d}$ denote the weight matrix, $\mathbf{x} \in \mathbb{R}^d$ the previous layer's post-activation, $\mathbf{b} \in \mathbb{R}^N$ the biases associated with the current layer. Moreover, let's define $\mathbf{y} \in \mathbb{R}^N$ as the vector of post-activations (activations after the ReLU), and $\tilde{\mathbf{y}} \in \mathbb{R}^N$ as the vector of pre-activations (activations before the ReLU). We can express the post-activations as:

$$\mathbf{y} = \max(\tilde{\mathbf{y}}, 0) = \max(\mathbf{W}\mathbf{x} + b, 0)$$

where

$$\mathbf{W} = \begin{bmatrix} \mathbf{w}_1^T \\ \mathbf{w}_2^T \\ \vdots \\ \mathbf{w}_n^T \end{bmatrix} \quad \mathbf{b} = \begin{bmatrix} b_1 \\ b_2 \\ \vdots \\ b_n \end{bmatrix}$$

Our goal is to find a colliding example that has the same post-activations as a target example. We can identify examples by their post-activations in the previous layer, since two examples with identical post-activations in the previous layer will always have identical pre- and post-activations in the current layer. We will denote the colliding example as $\mathbf{x}^*$ and the target example as $\mathbf{x}^t$ and define $\mathbf{y}^*, \tilde{\mathbf{y}}^*, \mathbf{y}^t, \tilde{\mathbf{y}}^t$ analogously.

Since ReLUs map all non-positive values to zeros, if the target example has a post-activation of zero in one component, as long as the pre-activation of the colliding example is non-positive in that component, then the target and the colliding example would have the same post-activation in that component. In order to make the post-activations of the two examples identical in all components, the following conditions are necessary and sufficient:

$$\forall i \text{ s.t. } \tilde{\mathbf{y}}_i^t > 0, \tilde{\mathbf{y}}_i^* = \mathbf{w}_i^T \mathbf{x}^* + b_i = \tilde{\mathbf{y}}_i^t$$
$$\forall i \text{ s.t. } \tilde{\mathbf{y}}_i^t \leq 0, \tilde{\mathbf{y}}_i^* = \mathbf{w}_i^T \mathbf{x}^* + b_i \leq 0$$

Consider the set of possible $\mathbf{x}^*$'s that would satisfy all these constraints. This set would be the intersection of the following sets:

$$\{\mathbf{x}^* : \mathbf{w}_i^T \mathbf{x}^* + b_i = \tilde{\mathbf{y}}_i^t\} \text{ for } i \text{ s.t. } \tilde{\mathbf{y}}_i^t > 0$$
$$\{\mathbf{x}^* : \mathbf{w}_i^T \mathbf{x}^* + b_i \leq 0\} \text{ for } i \text{ s.t. } \tilde{\mathbf{y}}_i^t \leq 0$$

Geometrically, each set that corresponds to an equality constraint represents a hyperplane in $d - 1$-dimensional subspace. Each set that corresponds to an inequality constraint represents a half-space. The intersection of the equality constraints is a subspace of at least $d - n_p$ dimensions, where $n_p$ is the number of components in $\tilde{\mathbf{y}}^t$ that are positive, assuming $d > n_p$. We consider the projections of the half-spaces onto this subspace, which are half-spaces in the subspace, and think about the intersection of all these half-spaces. In general, the intersection of finitely many half-spaces is a convex (but possibly unbounded) polytope. So, the intersection of all the sets listed above is a convex polytope in at least a $d - n_p$-dimensional subspace.

Any point in this polytope corresponds to a colliding example. Since there could be infinitely many points in a polytope, there could be infinitely colliding examples. If the polytope is bounded, we

---

[1] It should noted that the term "feature collision" was also used by (Shafahi et al., 2018) to refer to a different, but related, concept. In their context, feature collisions refer to input examples that are *both* close to a particular data example in input space and close to a different data example in feature space. In our context, input examples do not need to be close to any particular data example in input space and are only required to be unambiguously different from the target class to the human eye.

would be able to characterize all such examples if we know all the corners of the polytope, in which case the polytope would simply be the convex hull of all the corners. Then, any colliding example can be written as a convex combination of the corners, and we can generate a new colliding example by taking an arbitrary convex combination of the corners. (If the polytope is unbounded, we can still generate new colliding examples this way, but there could be colliding examples that are not convex combinations of the corners.)

Therefore, to find the set of colliding examples, we would need to find the corners of the polytope. To this end, we can move the dividing hyperplane of a half-space towards the feasible direction, which mathematically corresponds to decreasing the RHS of the corresponding inequality constraint. This is equivalent to picking a unit in the current layer and trying to make it as negative as possible. This process is illustrated in Figure 1 in 2D. To find a different corner, we simply pick a different constraint to optimize.

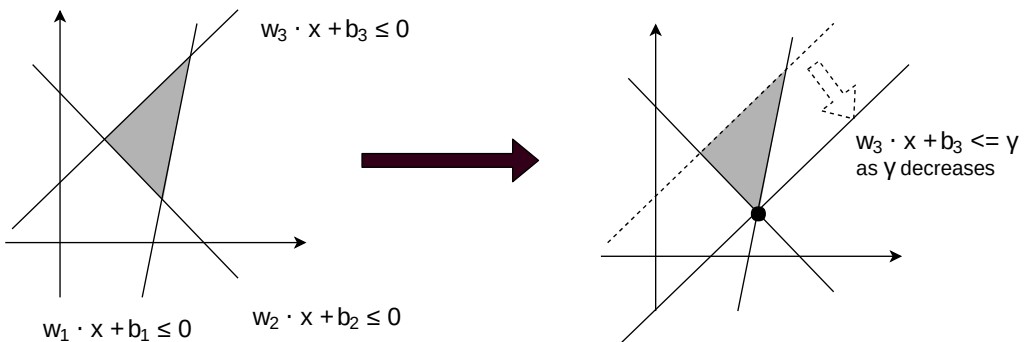

Figure 1: On the left, we illustrate the polytope that arises when there are three half spaces in a two-dimensional subspace. On the right, we illustrate how we can find a corner of a polytope.

In most of the literature on adversarial examples (Szegedy et al., 2013; Shafahi et al., 2018), the method of gradient descent, shown in Algorithm 1, is used to find these examples. Let $f(\mathbf{x})$ denote the neural network feature representation of input $\mathbf{x}$, which is the output feature of a certain layer. Given a target $\mathbf{t}$, generated feature collision sample $\mathbf{p}$ could be as simple as

$$\mathbf{p} = \arg\min_{\mathbf{x}} \mathcal{L}(\mathbf{x}) = \arg\min_{\mathbf{x}} ||f(\mathbf{x}) - f(\mathbf{t})||_2 + \mathcal{R}(\mathbf{x}) \qquad (1)$$

Here $\mathcal{R}(\cdot)$ is an application-specific penalization term that encourages the optimizer to find an example with some desirable property. Note that even though this method could find one such example, the solution would just be some arbitrary point in the polytope which is not helpful for exploring the full space of feature collisions.

---

**Algorithm 1** Gradient descent method.

**Given** a starting input $\mathbf{x}$ and learning rate $t$
**repeat**
    1. $\Delta\mathbf{x} := -\nabla\mathcal{L}(\mathbf{x})$.
    2. *Update.* $\mathbf{x} := \mathbf{x} + t\Delta\mathbf{x}$
**until** stopping criterion is satisfied.

---

Note that $f(\mathbf{x})$ is a normal feature output of a layer with ReLU activation, and we further define $f_0(\mathbf{x})$ to be the feature output before ReLU activation. To find the corners of polytope using the process in Figure 1 and yet still make use of the gradient-based method, we define two more loss term $\mathcal{L}_{pos}$ and $\mathcal{L}_{neg}$ where

$$\mathcal{L}_{pos}(\mathbf{x}) = ||f_0(\mathbf{x}) \odot \mathrm{psgn}(f_0(\mathbf{t})) - f(\mathbf{t})||_2$$

$$\mathcal{L}_{neg}^i(\mathbf{x}) = -(f_0(\mathbf{x}) \odot \mathrm{nsgn}(f_0(\mathbf{t})))_i + \sum_{j \neq i} |(f_0(\mathbf{x}) \odot \mathrm{nsgn}(f_0(\mathbf{t})))_j| \qquad (2)$$

$$j = 1, 2, \cdots$$

Here psgn($\cdot$) and nsgn($\cdot$) are two element-wise sign functions

$$\text{psgn}(x) = \max(\text{sgn}(x), 0) = \begin{cases} 0 & x < 0 \\ 1 & x >= 0 \end{cases}$$

$$\text{nsgn}(x) = \min(\text{sgn}(x), 0) = \begin{cases} -1 & x < 0 \\ 0 & x >= 0 \end{cases}$$

And we could update Equation 1 to be

$$p^i = \underset{\mathbf{x}}{\text{argmin}} \, \alpha \mathcal{L}_{pos}(\mathbf{x}) + \beta \mathcal{L}_{neg}^i(\mathbf{x}) + \mathcal{R}(\mathbf{x}) \tag{3}$$

where $i$ is an arbitrary index we choose to find one of the corners of the polytope.

Basically $\mathcal{L}_{pos}$ aims to keep the feature output after ReLU activation to be close to the target, as we only consider the units which are non-negative in the target feature. And for the negative units, if we assume that the feature output is a 1-D array, the first term of $\mathcal{L}_{neg}^i(\mathbf{x})$ would keep the selected unit to go smaller and the second term simply keeps other units close to zero. Since all influenced units are negative, and hence should be "wiped out" by ReLU layer, keeping them smaller or close to zero won't affect post-activation features after ReLU but would explore different corners of the polytope when choosing different units. For feature output more than 1 dimension, we could still do the same or choose a subset of the output units instead of just one.

## 3 EXPERIMENT

In this section we will apply our method to two different neural nets on MNIST and ImageNet dataset. Note that $\mathcal{R}(\mathbf{x}) = 0$ in this setting and we will directly optimize over the input image pixels. The loss function will be

$$\mathcal{L}^i(\mathbf{x}) = \alpha \mathcal{L}_{pos}(\mathbf{x}) + \beta \mathcal{L}_{neg}^i(\mathbf{x}) \tag{4}$$

### 3.1 MNIST DATASET

First we will try a simple two 256-unit hidden-layer fully-connected neural network on MNIST dataset (Damien (2017)).Both hidden-layers are followed by a ReLU activation. The model is trained to have a test accuracy of 96.64%.

Now, we use the first hidden layer output as our target feature representation and apply optimization with a fixed learning rate. An example is shown in Figure 2 where we start from a "3" and try to generate a feature collision with a "7". We do 5 optimization processes simultaneously and all settings are the same except $\mathcal{L}_{neg}$. Since we have 256 units on this layer, according to Equation 2 we use $\mathcal{L}_{neg}^0(\mathbf{x}), \cdots, \mathcal{L}_{neg}^4(\mathbf{x})$ for 5 optimization process respectively. We end the optimization after all $\mathcal{L}_{pos}$ is below 5 which only takes less than 30 minutes using a i7-4930K CPU.

In Table 1 we also show that the distance between generated samples are about 10% of normal distance of two different MNIST images. Although it is hard for human eyes to tell the difference, according to the numbers they are already different. Finally we take 2000 random interpolations of these generated samples, that is, a linear combination of the images using random weights. And as shown in Table 1, the top class is 100% correct (which is "7") and top class probability is the same as the target.

### 3.2 IMAGENET DATASET

And now we perform the same experiment on Imagenet, which includes more images and more complicated patterns. We now use a pre-trained VGG-16 net (Simonyan & Zisserman (2014)) that achieves 92.7% top-5 test accuracy on ImageNet. We use FC-6, which is the first fully-connected layer as our target feature output layer. There are 4096 units on FC-6 and we still use $\mathcal{L}_{neg}^0(\mathbf{x})$, $\cdots, \mathcal{L}_{neg}^4(\mathbf{x})$ for 5 optimization process respectively. The optimization process is very similar to the MNIST version. A fixed learning rate is applied and optimization continues until all $\mathcal{L}_{pos}$ is below a selected threshold. We have presented two cases in Figure 3 and the number detail is in Table 2. As the distance between generated samples is already very big, it is still hard for human eyes to notice the difference. Note that we still did 2000 random interpolations using the 5 generated samples and in both cases we achieve 100% class label accuracy. The top class probability also perfectly matches the target example.

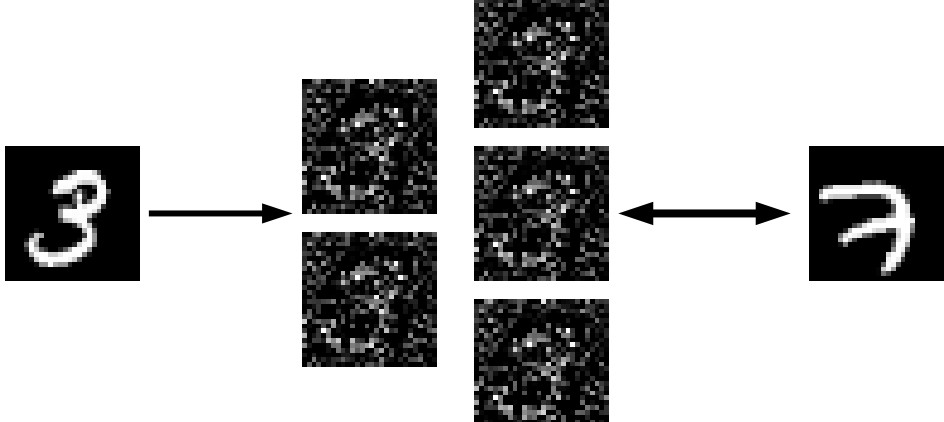

Figure 2: Mnist experiment result. Left is the initialization. 5 images in the middle are 5 generated results and they all share feature collision with the image on the right

| Average distance between images | |
| --- | --- |
| MNIST images | Optimized samples |
| 10.21 | 1.28 |

| Top class accuracy | Top class probability | |
| --- | --- | --- |
| | Target example | Interpolated samples |
| 100% | 1.0 | 1.0 |

Table 1: Details of MNIST case shown in Figure 2. Upper table shows the average $l_2$ distance between images from MNIST dataset and between images from the 5 optimized samples. The lower table shows the class accuracy (if it is a "7") of 2000 random interpolation of 5 generated samples and their top class probabilities.

## 4 MICRO-LEVEL VS MACRO-LEVEL DIFFERENCE

As many gradient-based methods to optimize the input image normally would result in noisy images, even if we have found different feature collisions using the above method, it would still be hard for human eyes to tell the difference unless paying close attention. This happens even when the distance between two images are far because the generated textures are irregular (i.e. noise) and

| | Average distance between images | |
| --- | --- | --- |
| | ImageNet images | Optimized samples |
| Figure 3(a) | 37538.22 | 4906.56 |
| Figure 3(b) | | 8959.69 |

| | Top class accuracy | Top class probability | |
| --- | --- | --- | --- |
| | | Target example | Interpolated samples |
| Figure 3(a) | 100% | 0.993 | 0.995 |
| Figure 3(b) | 100% | 0.999 | 0.999 |

Table 2: Details of ImageNet case shown in Figure 3. Upper table shows the average $l_2$ distance betweenimages from ImageNet dataset and between images from the 5 optimized samples. The Lower table shows the class accuracy of 2000 random interpolation of 5 generated samples for each case and the top class probabilities

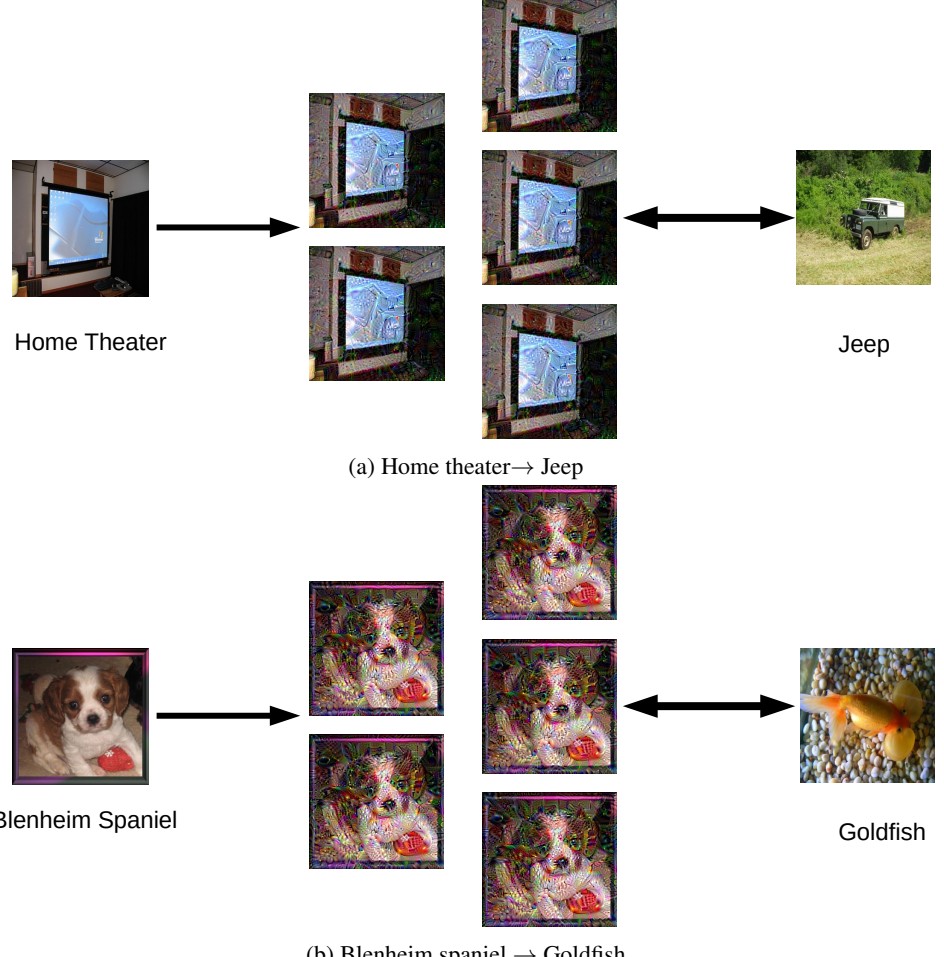

(a) Home theater→ Jeep

(b) Blenheim spaniel → Goldfish

Figure 3: Imagenet experiment result. (a) shows samples generated from a home theater that share feature collision with a jeep. (b) shows samples generated from a Blenheim Spaniel that share feature collision with a goldfish.

look confusing. Here we will call this micro-level difference. On the other hand, it would be very helpful to have human-recognizable differences, we will call it macro-level differences.

To get macro-level difference, obviously the images are expected to have more "natural" changes. So instead of directly optimizing the pixel value, we would want our generated pixels to be convex combination of "natura" pixels. To achieve this, instead of using one natural image, our initial image should be various natural source images which, when combined together, look natural. The source images could come from the training set where we search for images that are close to a seed image by using KNN. However KNN results are easily affected by dominant colors, meaning that we may not be able to get the details right. So we instead extract smaller patches from the seed image and combine the nearest neighbors of those patches to get our initial image. Due to the large size of modern datasets and the huge number of queries, KNN could be very expensive in terms of time and space. However thanks to recent advances in nearest neighbour search algorithms (Li & Malik, 2017), this problem is substantially alleviated.

The initialization process is as follows: we first extract patches using a sliding window, of which the size is our designed patch size $n \times n$. This could also be done using a convolutional layer where there is only 1 $n \times n$ filter with all elements being 1. Assume that we have $m$ patches, then by searching for nearest neighbors of these patches in the dataset, $m$ nearest neighbors will be found. For each nearest neighbor, as shown in Figure 4, we paste the nearest neighbor as well as fixed size of surroundings at the same position of the query patch on a blank image. The surroundings are for interpolation use.

Let $s$ denote the size of the surroundings. A more detailed source image structure is shown in Figure 5. Now let $Q = \{q^0, q^1, \cdots, q^{m-1}\}$ denote the source images and $C = \{(c_x^0, c_y^0), \cdots, (c_x^{m-1}, c_y^{m-1})\}$ denote the center coordinates of the patches in source images. We combine the source images together with linear interpolation to generate the composite image $I$, where

$$I_{i,j} = \frac{\sum_{k \in A_{i,j}} ((n+2s)/2 - |i - c_x^k|)((n+2s)/2 - |j - c_y^k|) q_{i,j}^k}{\sum_{k \in A_{i,j}} ((n+2s)/2 - |i - c_x^k|)((n+2s)/2 - |j - c_y^k|)}$$

Here $A_{i,j}$ denote the set of all source images of which the pasted area includes position $(i, j)$, that is

$$A_{i,j} = \{k | (n+2s)/2 - |i - c_x^k| \geq 0, (n+2s)/2 - |j - c_y^k| \geq 0\}$$

We can further write

$$I_{i,j} = \sum_{k \in A_{i,j}} p_{i,j}^k q_{i,j}^k$$

where

$$p_{i,j}^k = \frac{((n+2s)/2 - |i - c_x^k|)((n+2s)/2 - |j - c_y^k|)}{\sum_{k \in A_{i,j}} ((n+2s)/2 - |i - c_x^k|)((n+2s)/2 - |j - c_y^k|)} \quad (5)$$

is called the *control parameter* for the $k$th source image at position $(i, j)$. Obviously, the control parameters have the same size as the source images, and generating a composite image only takes an element-wise multiplication of the two tensors and an addition over all images.

Now instead of updating the input image for each optimizing step, we update the control parameters. Define the composite image as

$$\mathbf{x} = G(P, Q)$$

where $P$ is the control parameter tensor and $Q$ is the source image tensor, $G$ is the process described above. The loss function $w.r.t\ P$ is

$$\mathcal{L}^i(P) = \alpha \mathcal{L}_{pos}(G(P, Q)) + \beta \mathcal{L}_{neg}^i(G(P, Q)) + \mathcal{R}(P)$$

And the optimization algorithm is the same as Algorithm 1 except the gradient becomes

$$\frac{\partial \mathcal{L}}{\partial P} = \frac{\partial \mathcal{L}}{\partial \mathbf{x}} \cdot \frac{\partial \mathbf{x}}{\partial P} = \nabla \mathcal{L}(\mathbf{x}) \cdot \nabla G_P$$

The entire process is shown in Figure 6

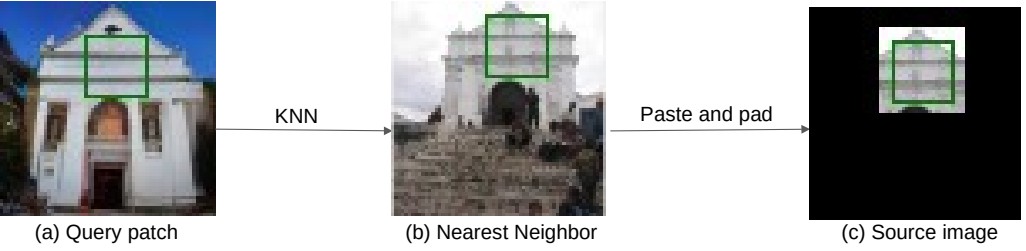

(a) Query patch     (b) Nearest Neighbor     (c) Source image

Figure 4: This figure shows how we get source images. (a) is the query image and the green rectangle labels a query patch. Using KNN, we get the nearest neighbor, which is also labeled by green rectangle in (b). (c) is the source image where on a blank image, the patch is pasted on the same position of the query patch. Fixed size of surroundings are also pasted for interpolation use.

## 4.1 PENALTY TERM

To reveal the macro-level change described in 4, the generated images should be more natural and smooth. To achieve this, we use a penalty term that penalizes the difference between weights on adjacent pixels of the same source image. If there are $K$ source images of size $M \times N$, that is

$$\mathcal{R}(P) = \sum_{k=1}^{K} (\sum_{i=1}^{M} \sum_{j=1}^{N-1} |p_{i,j}^k - p_{i,j+1}^k| + \sum_{j=1}^{N} \sum_{i=1}^{M-1} |p_{i,j}^k - p_{i+1,j}^k|)$$

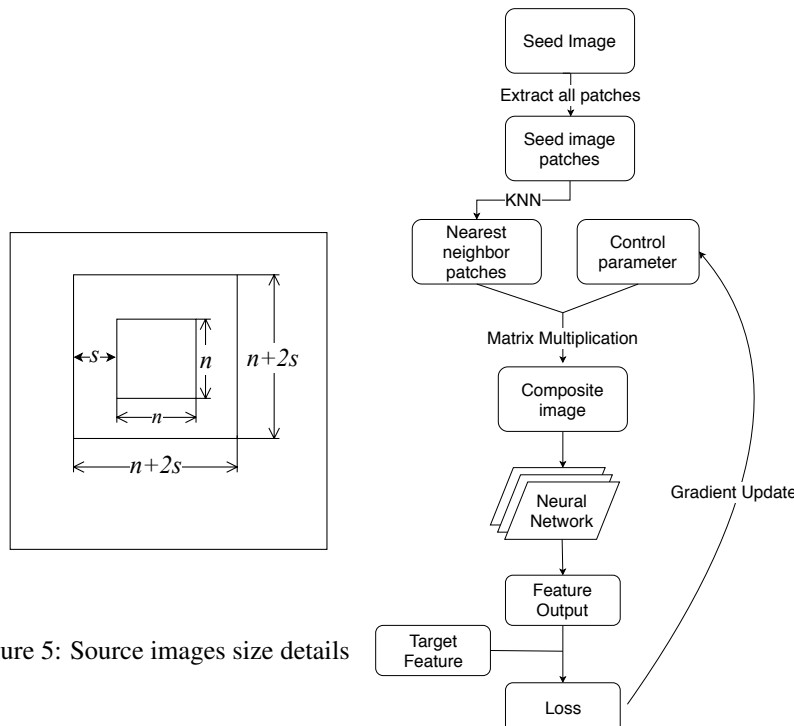

Figure 5: Source images size details

Figure 6: A diagram about how to generate samples with macro-level difference

And the loss function would be

$$\mathcal{L}^i(P) = \alpha\mathcal{L}_{pos}(G(P,Q)) + \beta\mathcal{L}_{neg}^i(G(P,Q)) + \gamma\mathcal{R}(P) \tag{6}$$

## 4.2 MACRO-LEVEL DIFFERENCE RESULT

Now as stated above, we would like to generate samples that share macro-level differences. We still apply this to VGG-16 net with ImageNet dataset.

To keep images natural we will use the penalty term described in Section 4.1. And the loss function will be Equation 6 and we still choose $\mathcal{L}_{neg}^0(\mathbf{x})$, $\cdots$, $\mathcal{L}_{neg}^4(\mathbf{x})$ respectively for 5 optimization processes. To do KNN search, we first generate a patch dataset to search from. We randomly select 10% images from each class and extract patches using a sliding window with constant pixel strides along horizontal and vertical directions. We ended up with a dataset with $37011074$ $32 \times 32$ patches from all $1000$ classes of ImageNet. Given a query patch, the Prioritized DCI-KNN (Li & Malik (2017)) could find the nearest neighbor in 5 minutes.

In our setting, starting from a seed image, we first extract patches also using a sliding window. After doing KNN search, we generate the relative control parameters. Then following the process described in Figure 6, we can start optimizing the control parameters. Note that in the experiment, we use softmax to make sure for each pixel every control parameter is in the range (0, 1) and add up to 1, which is a little different from what is described in Equation 5. Also, an increasing learning rate will be used to accelerate optimization.

In Figure 7 we have shown results of 2 experiments. The five generated samples are different from each other in a more obvious way. And yet the random interpolation still reaches 100% class accuracy. Interestingly, in Figure 7(b) the target's real class is a carton and is misclassified as an eraser. Since we are matching the feature representation of the initialization and the target, generated samples and their interpolations are all classified as "eraser".

In addition, the reason why these macro-level difference exists is because by KNN, we gathered various kinds of source patches. For some query patch, when using pixel-wise euclidean distance as metric, it is likely that we will end up with a nearest neighbor that looks dissimilar from the query.

For example, a patch filled with black and white pixels could end up with a patch with only gray pixels. Even though when combined together we won't notice such big difference, after optimization, control parameters might reveal these patches and thus resulting in the macro-level differences in Figure 7.

As shown in Table 3, the optimized samples could share similar $l_2$ distances with each other as Table 2, while at the same time there are more obvious and natural differences. Top class probability keeps to be close to the target example. Therefore, it proves that potential feature collisions take over a huge space and various images exist in between.

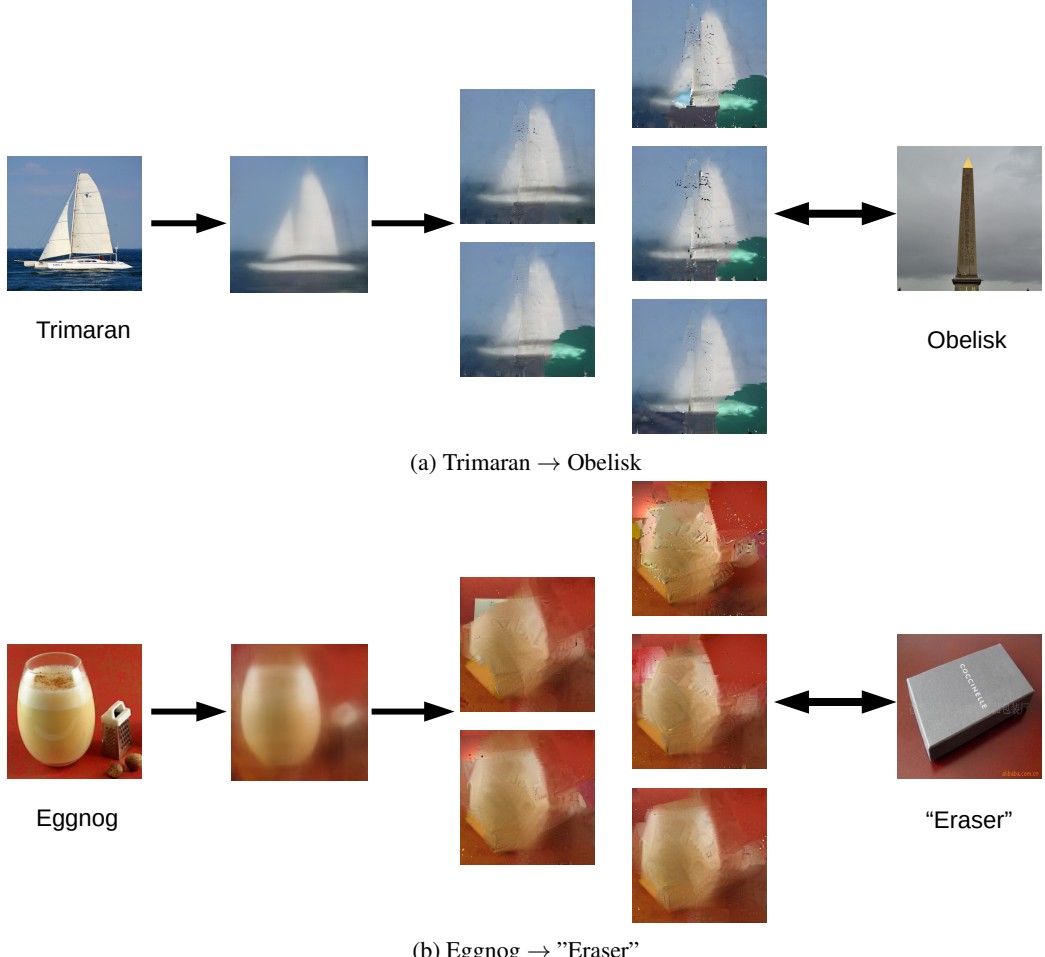

(a) Trimaran → Obelisk

(b) Eggnog → "Eraser"

Figure 7: Imagenet experiment result. (a) shows samples generated from a trimaran that share feature collision with an obelisk. (b) shows samples generated from a eggnog that share feature collision with a misclassified "eraser".

## 5 RELATED WORK

There have been several lines of work that use iterative optimization to find noteworthy input examples. One line of work is on adversarial examples, where the goal is to find a small perturbation to a source input example so that it is misclassified. Tatu et al. (2011) proposed using projected gradient descent to find a perturbed version of an example with similar SIFT features as an example from a different class. Szegedy et al. (2013) demonstrated a similar phenomenon in neural nets, where an adversarially perturbed example can be made to be classified by neural nets as an example from any arbitrarily class. Nguyen et al. (2015) further shows that it is easy to generate images that do not resemble any class, but are classified as a recognizable object with high confidence. Kurakin et al.

| | Average distance between images | |
|---|---|---|
| | Imagenet images | Optimized samples |
| Figure 7(a) | 37538.22 | 5909.37 |
| Figure 7(b) | | 6074.86 |

| | Top class accuracy | Top class probability | |
|---|---|---|---|
| | | Target example | Interpolated samples |
| Figure 7(a) | 100% | 0.997 | 0.992 |
| Figure 7(b) | 100% | 0.579 | 0.585 |

Table 3: Details of Imagenet case shown in Figure 7.

(2016) demonstrated that even when adversarial examples are printed on a sheet of paper, they are still effective at fooling neural nets. Athalye & Sutskever (2017) even managed to 3D print adversarial example models. More recently, Shafahi et al. (2018) showed that it is possible to adversarially perturb examples so that their features are close to the features of a completely different example.

Similar techniques have also been used for a different purpose, namely to understand what input image would cause the activation of a particular neuron in neural nets to be high, which is known as *activation maximization* (Erhan et al., 2009). This technique can be applied to either a neuron in the output layer (Simonyan et al., 2013) or a hidden layer (Erhan et al., 2009; Yosinski et al., 2015). A related line of work, known as *code inversion* (Mahendran & Vedaldi, 2015; Dosovitskiy & Brox, 2016), aims to find an input image whose entire activation vector after a particular layer is similar to the activation vector of a particular real image. Unlike the adversarial example literature, the goal of this body of work is to find an *interpretable/visually recognizable* image that allows for the visualization of the kinds of images that would either cause a particular neuron to activate or all neurons in the same layer to exhibit a particular pattern. Therefore, a regularizer is usually included in the objective function that favours images that are more natural, e.g.: those that are smooth and do not have high frequencies. Surprisingly, the images that are found often bear resemblance to instances in the target class. It has been conjectured (Mahendran & Vedaldi, 2015; Nguyen et al., 2016) that the inclusion of this regularizer would explain the apparent discrepancy between the findings of the adversarial example literature and the code inversion literature – code inversion is simply not finding adversarial examples because having high-frequency perturbations is penalized by the regularizer. Our experiments show that this may in fact not be true, since we are able to successfully find examples that do not have high frequencies but clearly do not resemble any instance of the target class. We conjecture this may be because points near the centre of the polytope may not be reachable using a naïve loss function that only penalizes the difference between post-activations; this is because the gradient becomes zero as soon as the input example is moved into the polytope. As a result, the solution found using gradient descent will usually be near the boundary of the polytope, which may happen to resemble objects in the target class.

## 6 CONCLUSION

In this paper, we have shown theoretically that polytopes of misclassified examples could exist due to the properties of ReLU activation functions. We developed a method for finding such polytopes and demonstrated empirically that they do in fact exist in commonly used neural nets. Somewhat surprisingly, in Section 4.2, we demonstrated that even after constraining examples to be compositions of image patches, these polytopes still exist. Furthermore, the corners of the polytope appear perceptibly different, which shows that interpolations in this polytope can all be misclassified even though each interpolation is visually distinctive. While non-linearities in neural nets are crucial to ensure high expressive power Cybenko (1989), we show that they could also cause neural nets to be overly robust to *all* examples in certain polytopes, which is an important trade-off to bring to light.

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

## APPENDIX A    MORE MACRO-LEVEL EXPERIMENTS

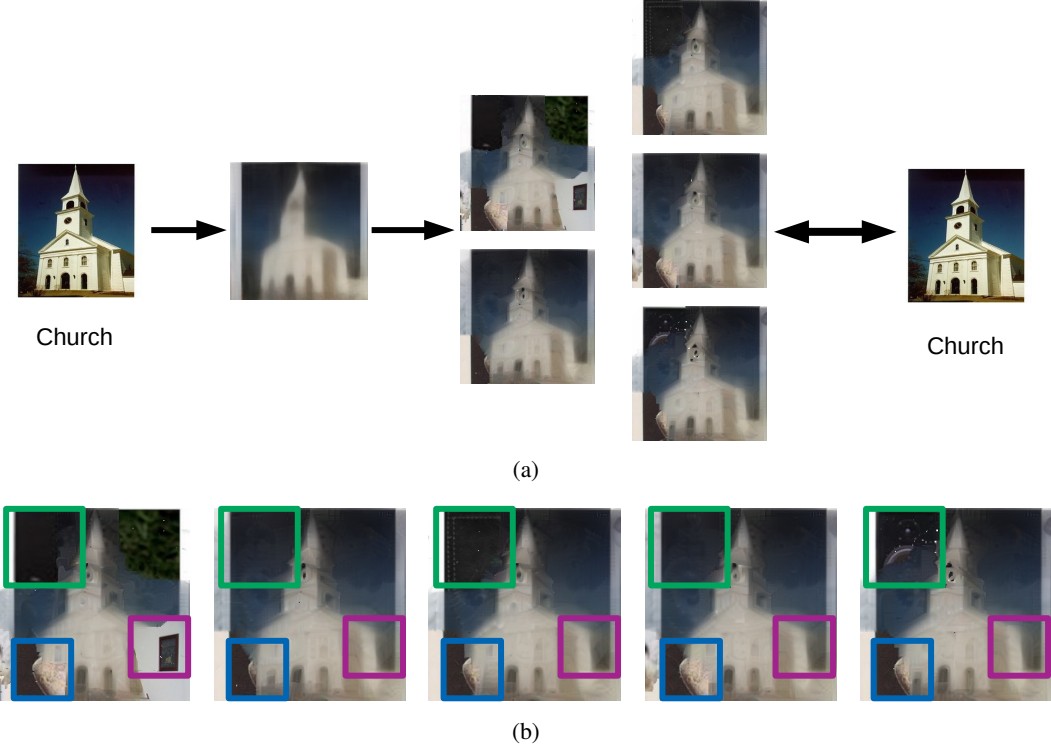

Figure 8: (a) ImageNet result that shows five corners of a polytope, within which all examples collide with the original class (church). (b) We highlight the differences between the corners of the polytope using different coloured boxes. For example, in the region enclosed by the green boxes, the third image has dotted lines and the fifth image has a white arc. In the region enclosed by purple boxes, the first image has a square, which does not exist in other images. In the region enclosed by blue boxes, the first and fourth images have a large beige blob, whereas other images have much smaller blobs.

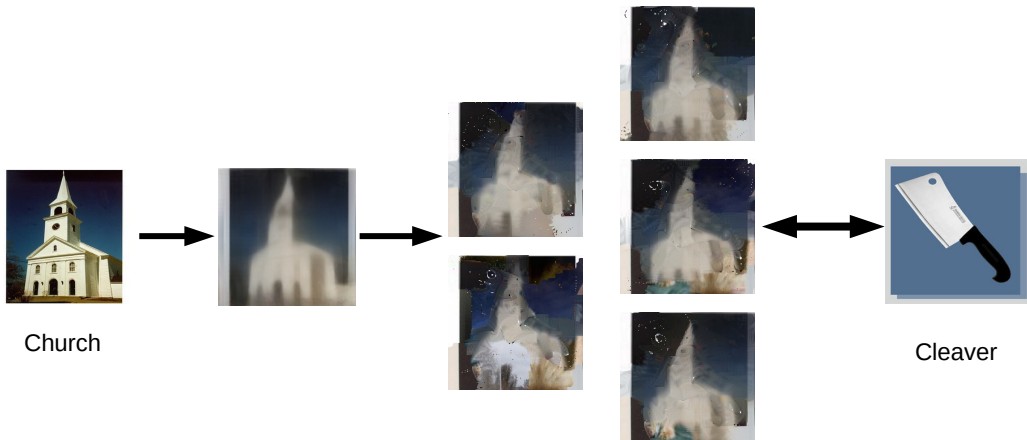

Figure 9: ImageNet result that shows five corners of a polytope, within which all examples collide with a different class (cleaver).

| | Average Pairwise Distance between Images | |
|---|---|---|
| | ImageNet Images | Polytope Corners |
| Figure 8 | | 8597.34 |
| Figure 9 | 37538.22 | 9587.92 |

| | Top Class Accuracy | Top Class Probability | |
|---|---|---|---|
| | | Target Example | Samples from the Polytope |
| Figure 8 | 100% | 0.952 | 0.877 |
| Figure 9 | 100% | 0.999 | 0.8 |

Table 4: Quantitative results corresponding to Figures 8 and 9. The top table shows that the average distance between different corners of the polytope is not much less than the average distance between two random images from the dataset (ImageNet), which demonstrates that the polytope is not small. The bottom table shows that *all* 2000 samples from the polytope are classified as/collide with the target class.

## APPENDIX B    COMPARISON OF POLYTOPE AND BALL

To demonstrate that the polytope we find cannot be trivially replaced with a ball of similar size, we consider a ball centred at the centroid of the polytope whose radius is the minimum distance from the centroid to a corner of the polytope. We then randomly sample points inside the ball and check whether they are classified as/collide with the target class. The percentage of points that collide with the target class is shown in Table 5. Whereas the percentage of points inside the polytope that collide with the target class is 100%, the percentage of points inside the ball that collide with the target class is much smaller, as shown. This demonstrates that samples from a ball of similar size does not always collide with the target class, unlike in the case of the polytope.

| | Top class accuracy |
|---|---|
| Figure 7(a) | 24.8% |
| Figure 7(b) | 3.5% |
| Figure 8 | 9.2% |
| Figure 9 | 0.0% |

Table 5: Percentage of points from a similarly sized ball centred at the centroid of the polytope that are classified as the target class.

## APPENDIX C    INTERMEDIATE IMAGES DURING OPTIMIZATION

To better understand the optimization process of finding the polytope we visualize the intermediate images during optimization. Figures 10, 11 and 12 each show the intermediate images at different points of the optimization process for two corners randomly chosen from the five that are shown in Figures 7(a), 7(b) and 9 respectively.

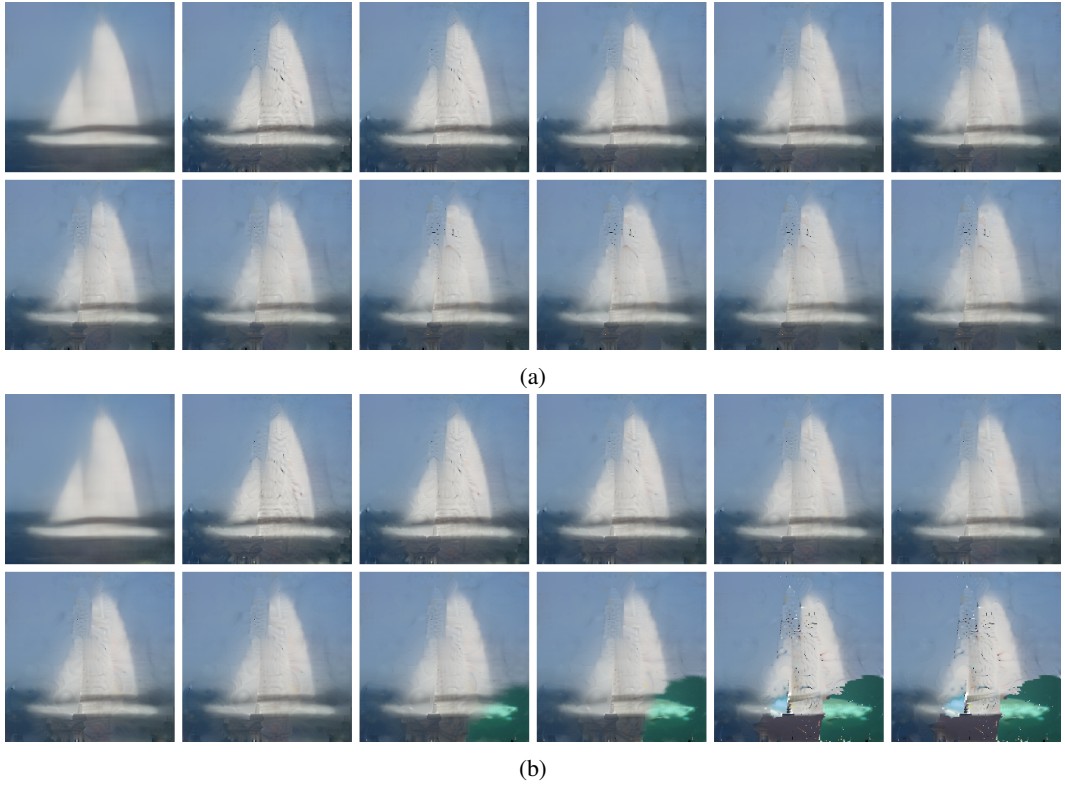

Figure 10: Intermediate images for Figure 7(a)

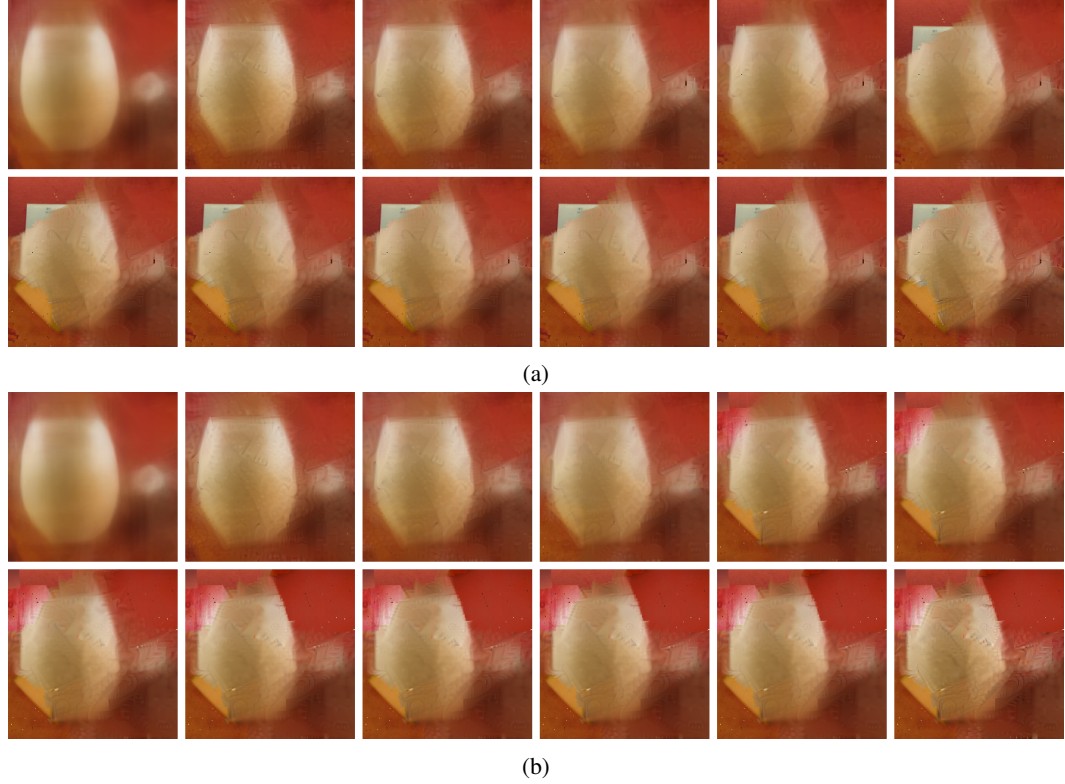

Figure 11: Intermediate images for Figure 7(b)

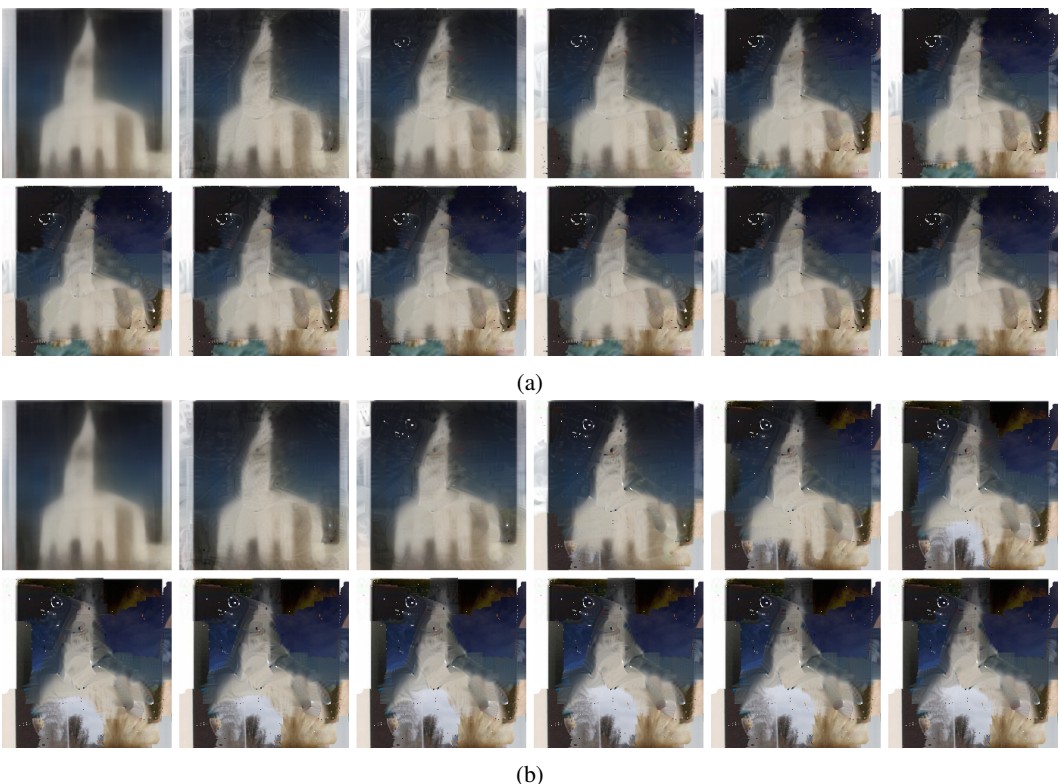

Figure 12: Intermediate images for Figure 9

