# OpenReview forum: "A Study of Robustness of Neural Nets Using Approximate Feature Collisions"
_ICLR.cc/2019/Conference_

### Official Review · AnonReviewer1 · 2018-11-01
**Interesting premise of adversarial polytopes, but fall on implication and attempt to create macro-level different examples.**

**Rating:** 4
**Confidence:** 4

**Review:**

This paper presents an algorithm for finding a polytope of adversarial examples. This means that within a convex hull, you can move around freely and get a new adversarial example at each point, while still maintaining misclassification. It then couples this with a method of generating nearest neighbor patch-based images in an effort to create "macro-level different" examples. The premise is interesting, but the implications are questionable and I do not find the work in macro-level differences to be sound. This could be based in misunderstandings, so please let me know if you think that is the case.

Strengths:
- The notion of the polytope is interesting and the algorithm for finding such polytope seems perfectly reasonable.
- I think the goal of macro-level adversarial examples is interesting.

Weaknesses:
- First of all, the 5 corners of the polytope all look the same to me (for instance fig 2). This is not encouraging, because it means that every single point in the polytope will also look exactly like the corners. To be frank, this means the polytope is not that interesting and has only found an extremely small pocket of adversarial examples. If you use a regular method of finding a single adversarial example, I'm sure the outcome wouldn't change within some ball around the sample (perhaps with very small radius, but nonetheless). In fact, a comparison between that ball's volume and the volume of the polytope would be interesting.
- The implication of these polytopes is not at all clear if it doesn't really allow us to generate adversarial example of a new flavor. The investigation into macro-level differences does not help the case, as I will explain.
- I am not at all convinced that there is any meaning to the examples with "macro-level differences."  It's a bit unclear to me how many patches are used per image, but assuming that a patch is centered over each pixel,  it would mean that we have as many control parameters as we have pixels, which assuming the pixels each have three color values, is just 1/3 of the original degrees of freedoms. Now, the patches probably do constrain what we can paint a bit, but since the patches are applied with a pyramid, it means the center pixel will contribute more than any other for a given patch, so I'm not so sure. I'm not convinced that we can't come up with linear combinations of these patches that produce highly non-natural images with "micro-level" adversarial patterns. In fact, I think section 4.1 and figure 7 provide evidence to the contrary. Let me explain:
    - Section 4.1: Why do you need a total variation penalty at all if you have constructed a patch-based drawing method that is supposed to be unable to produce unnatural high-frequency patterns? If you only had a handful of patches and they were all non-overlapping, then this would be impressive and.
    - Figure 7: We can clearly see high-frequency patterns that create the shadow of an obelisk in 7(a). I think the same is true for "erase", although the pattern is not as recognizable. The examples actually look more suspicious than regular adversarial examples, since it looks like the original image has simply been blurred, which means the adversarial perturbations are more clear. I understand that these patterns were created using a complicated scheme of natural patches, but I think you made this method too powerful. The one interesting quality is the bottom right of the trimaran which looks like a shark - however, that is a singular occurrence in your examples and it certainly feels like the high-frequency patterns will contribute much more to class than the shark itself.
- Please let me know if I am misinterpreting the importance of the results in Figure 7, since this is an important culmination of this work.

Other comments:
- Some of notation is a bit confusing. In (1), why is p not bold but x and t are bold? They are all vectors. In Algorithm 1, x is not bold anymore.
- Algorithm 1 also seems quite unnecessary to include so explicitly.
- Isn't a bounded polytope called a "simplex"? Perhaps there is a distinction that I'm not aware of, but the absence of the word "simplex" throughout the whole paper surprised me a bit. Perhaps this is a perfectly correct omission due to differences that I'm not aware of.

Minor comments:
- abstract, "We propose a way to finding" -> either "to->"of" or "find"
- page 3, "and we can generate new colliding example" -> "a new colliding example"
- page 3, "taking arbitrary an convex combinations" -> "combination"
- page 3, "Given a target x", I think you mean "Given a target t"
- page 5, "As many gradient-based method" -> "methods"
- page 8, "carton"? "rubber"? Those are not in figure 7(b).
- page 10, "are crucial to less non-robust" ? This sentence (which is the final sentence of the conclusion and thus has a certain level of importance) is not something that is novel to your paper. The impact of non-linearities on adversarial examples have been well-studied.

---

> ### Author Response · Authors · 2018-11-28
> **Response to your review (1/2)**
>
> Thank you for your review. First, we would like to clarify some misunderstandings:
>
> The purpose of the paper is *not* to generate adversarial examples, but rather to demonstrate the existence of polytopes within which all examples share similar feature activations and therefore classifications. The paper demonstrates that such polytopes can be found in the neighbourhood of *any* example (that we tested) and can be made to be classified as *any* target class by the neural net. Now, the question is: why is this interesting?
>
> - This shows that a neural net can map an infinite number of examples to the same or similar feature activations. This may not be desirable, because the different examples in the polytope are visually different (especially under the macro-level setting), but the neural net is essentially “blind” to these changes in visual appearance because the feature activations and therefore the classifications don’t change.
> - This cannot be easily fixed by augmenting the training set because the polytopes arise from the properties of ReLU activation functions; training the neural net on a different training set can only make the weights of the neural net different and so cannot in general eliminate the existence of the polytopes.
>
> This is different from the adversarial example setting in two important ways:
> - The goal of adversarial examples is to find an example that is close to the initial image. Our goal is to find corners of the polytope that are as *far* from the initial image as possible. (In Tables 1, 2, 3 and 4, we show that the corners of the polytope are fairly far apart compared to the average pairwise distances between images from the dataset, which is desirable in our setting because this means the polytope is large.)
> - The goal of adversarial examples is to find an example that is classified as another category. Our goal is to find a polytope that results in feature activations that are the *same* as the feature activations of some target image. The latter is in some sense stronger, since matching feature activations will guarantee matching classifications, but not the other way around. It is also different from the goal of adversarial examples, since the target image could be chosen to be in the same class as the initial image. (We show such an example in Figure 8.) This would not be a problem from the perspective of adversarial examples, but is still undesirable, because the target image can be visually very different from the initial image, but the feature activations are very similar.
>
> Below are our responses to your questions and concerns:
>
> Q1: First of all, the 5 corners of the polytope all look the same to me ... this means the polytope is not that interesting and has only found an extremely small pocket of adversarial examples.
> A1: It’s important to distinguish between two distinct questions: whether the polytope is small and whether the different examples in the polytope are perceptibly different. In Figure 3 for example, the average pairwise distance between the corners of the polytope is 1/4 to 1/8 of the average pairwise distance between images from ImageNet (as shown in Table 2), and so the polytope is not small. One can argue, however, that the corners in Figure 3 are not perceptibly different, which is why we introduced a method for finding polytope corners that are visually different at the macro-level.
>
> Q2: If you use a regular method of finding a single adversarial example, I'm sure the outcome wouldn't change within some ball around the sample (perhaps with very small radius, but nonetheless). In fact, a comparison between that ball's volume and the volume of the polytope would be interesting.
> A2: As explained above, because our goal is *not* to find adversarial examples, we cannot compare to a regular adversarial example method. However, we have updated our paper to include a comparison in the appendix to a baseline that is similar to what you are suggesting in spirit, which is to use a ball centred at the centroid of the polytope that we find, whose radius is similar to the radius of the polytope. Specifically, we randomly select 2000 examples inside a ball centred at the centroid of our polytope whose radius is the minimum distance between the centroid and a corner of the polytope and compare the percentage of these examples that are classified as (or in other words, collide with) the target class. As shown in Table 5, only a small fraction of examples collide with the target class, compared to 100% success rate when drawing samples from the polytope. This demonstrates that the polytope we find is interesting and cannot be trivially replaced with a ball.
>
> (continued below)

---

> > ### Author Response · Authors · 2018-11-28
> > **Response to your review (2/2)**
> >
> > Q3: I'm not convinced that we can't come up with linear combinations of these patches that produce highly non-natural images with "micro-level" adversarial patterns ... Section 4.1: Why do you need a total variation penalty at all if you have constructed a patch-based drawing method that is supposed to be unable to produce unnatural high-frequency patterns?
> > A3: We did not make either of these claims in our paper - specifically, we made no claims about the impossibility of finding a linear combinations of patches to produce an arbitrary image, or about the utility of a patch-based parameterization without a total variation penalty. So, we are not sure why this is a criticism. In fact, it is obvious that it *is* possible to find a linear combination of patches to produce an arbitrary image (if given enough patches), which is a simple consequence of basic linear algebra. This is precisely why we are constraining the space of control parameters when performing optimization - we always enforce a *convex* combination of patches and use a regularizer to encourage spatial smoothness in the coefficients on the patches. We did not claim that having this regularizer is somehow undesirable or unnecessary; in fact, we very much designed this regularizer to go hand-in-hand with the patch-based parameterization.
> >
> > Q4: The examples actually look more suspicious than regular adversarial examples, since it looks like the original image has simply been blurred, which means the adversarial perturbations are more clear.
> > A4: As explained above, the goal of the paper is not to find adversarial examples - rather it is to find corners of a polytope that causes feature collisions with a target example, so that we can show such polytopes exist. Our goal is to find a *large* polytope that has *similar* feature activations, whereas the goal of adversarial examples is to find an image with *small* distortion that results in a *different* classification.
> >
> > Q5: Isn't a bounded polytope called a "simplex"? Perhaps there is a distinction that I'm not aware of, but the absence of the word "simplex" throughout the whole paper surprised me a bit. Perhaps this is a perfectly correct omission due to differences that I'm not aware of.
> > A5: A simplex in d-dimensional space can only have d+1 vertices (since they all have to be affinely independent), whereas a convex polytope doesn’t have this requirement. This means that a simplex is a bounded convex polytope, but not all bounded convex polytopes are simplices. Because any bounded convex polytope can be decomposed into simplices, the existence of a polytope implies the existence of a simplex, but the former is a stronger statement, which is why we talk about polytopes rather than simplices.
> >
> > Thanks for the fixes; we’ve updated our paper and incorporated them.

---

### Official Review · AnonReviewer2 · 2018-11-02
**Interesting but limited study**

**Rating:** 4
**Confidence:** 4

**Review:**

This paper follow recent trend of adversarial examples which is on generating images with small differences in the input space, but that are misclassified by a large margin by a neural net. The key idea of the paper is that any negative component before a ReLU activation share the same zero feature after the ReLU. Thus, any neural network that has ReLU activations have a polytope in the input space that will have identical activations in the later layers. Based on this observation, the paper assert that such polytope always exist and describe how to find its corners with a gradient descent based method. Two simple experiments on MNIST and ImageNet datasets are carried to show the feasibility of the method in practice and the existence of images with feature collision, together with their average L2 distance from real images. Since the images are clearly not "natural" images, a further method based on selecting patches of real images is reported and tested on ImageNet. This shows that the approach can be further applied on macro-level differences.

Strengths
+ The observation of the existence of the polytope in presence of ReLU activation is interesting and can probably be used to further refine attacks for generating adversarial examples.
+ The paper is clear and is comprehensive of all the basic steps.
+ Examplar experiments show the possibility of using the key idea to generate adversarial examples

Weaknesses:
- The experiments are very limited and show just 5 examples of generated images on MNIST and ImageNet. In Sect 3.2 it is observed that it is hard for human eyes to notice the difference but that is clearly not the case for the figure reported. The same for Fig. 7 on the macro-level which are even more distorted. Although this is minor, since the method is still shown to be working, the statements on the similarity of images seem incorrect. Beside the qualitative examples, the measurement of average similarity based on L2 is not so indicative at the perception level, but still interesting to see.
- No comparison with other methods to generate adversarial examples are reported (e.g. Shafani et al 2018, Szegedy et al. 2013).

Minor issues:
- Figure 2, Figure 3 show the results, but it would also be interesting to observe what happens from the starting image to the final generated images.
- Personally, I prefer to see related work after the introduction section. Reading it at the end breaks the flux of the paper.
- The observation is only applicable to ReLU activations (but other activation functions may be in the last layer), limiting the impact of the paper.

---

> ### Author Response · Authors · 2018-11-28
> **Response to your review**
>
> Thank you for your review. First we would like to clarify some misunderstandings:
>
> The purpose of the paper is *not* to generate adversarial examples, but rather to demonstrate the existence of polytopes within which all examples share similar feature activations and therefore classifications. The paper demonstrates that such polytopes can be found in the neighbourhood of *any* example (that we tested) and can be made to be classified as *any* target class by the neural net. Now, the question is: why is this interesting?
>
> - This shows that a neural net can map an infinite number of examples to the same or similar feature activations. This may not be desirable, because the different examples in the polytope are visually different (especially under the macro-level setting), but the neural net is essentially “blind” to these changes in visual appearance because the feature activations and therefore the classifications don’t change.
> - This cannot be easily fixed by augmenting the training set because the polytopes arise from the properties of ReLU activation functions; training the neural net on a different training set can only make the weights of the neural net different and so cannot in general eliminate the existence of the polytopes.
>
> This is different from the adversarial example setting in two important ways:
> - The goal of adversarial examples is to find an example that is close to the initial image. Our goal is to find corners of the polytope that are as *far* from the initial image as possible. (In Tables 1, 2, 3 and 4, we show that the corners of the polytope are fairly far apart compared to the average pairwise distances between images from the dataset, which is desirable in our setting because this means the polytope is large.)
> - The goal of adversarial examples is to find an example that is classified as another category. Our goal is to find a polytope that results in feature activations that are the *same* as the feature activations of some target image. The latter is in some sense stronger, since matching feature activations will guarantee matching classifications, but not the other way around. It is also different from the goal of adversarial examples, since the target image could be chosen to be in the same class as the initial image. (We show such an example in Figure 8.) This would not be a problem from the perspective of adversarial examples, but is still undesirable, because the target image can be visually very different from the initial image, but the feature activations are very similar.
>
> Below are our responses to your questions and concerns:
>
> Q1: The experiments are very limited and show just 5 examples of generated images on MNIST and ImageNet.
> A1: We have to show a limited number of images for reasons of space - showing more images would be somewhat boring and would not add much value to the paper, since the point of the paper was to show the existence of polytopes where features collide. The algorithm for finding such polytopes is simple enough for any reader to implement on their own and verify the existence of polytopes around images of their own choosing.
>
> Q2: In Sect 3.2 it is observed that it is hard for human eyes to notice the difference but that is clearly not the case for the figure reported. The same for Fig. 7 on the macro-level which are even more distorted.
> A2: As explained above, our goal is different from that of adversarial examples - the point is to *maximize* distortion, not to *minimize* distortion. Our goal is to find a *large* polytope that has *similar* feature activations, whereas the goal of adversarial examples is to find an image with *small* distortion that results in a *different* classification.
>
> Q3: No comparison with other methods to generate adversarial examples are reported (e.g. Shafani et al 2018, Szegedy et al. 2013).
> A3: As explained above, since our goal is *not* to generate adversarial examples, it would not be possible to compare to methods for generating adversarial examples.
>
> Q4: Figure 2, Figure 3 show the results, but it would also be interesting to observe what happens from the starting image to the final generated images.
> A4: We have added intermediate images in Appendix C for all three macro-level difference experiments.
>
> Q5: The observation is only applicable to ReLU activations (but other activation functions may be in the last layer), limiting the impact of the paper.
> A5: The ReLU activations do not have to be in the last layer for our method to work - as long as there are ReLU activations in *some* layer, then our method is applicable. In fact, in our paper, we used the first fully-connected layer as opposed to the last layer for finding feature collisions. The idea is that once the feature activations of an earlier layer collide, the feature activations of the following layers will collide as well. Because ReLU activations area quite common in neural net architectures, our method is broadly applicable.

---

### Official Review · AnonReviewer3 · 2018-11-05
**interesting observation and techniques, but results leave something to be desired**

**Rating:** 6
**Confidence:** 3

**Review:**

This paper studies a non-local form of adversarial perturbation, which, to my limited knowledge is new. The form of the perturbation is specific to ReLU activations, but it may be a large set. The authors also devise an algorithm to generate natural-looking perturbations in this set. Instead of updating a seed example through gradient descent, they propose to generate perturbations by combinations of image patches from multiple seed images. The weights of the combination are optimized by a gradient descent like algorithm in a similar manner as standard gradient-based approaches to generating adversarial examples. This produces perturbations that look like ```in-paintings'' or transplants of one seed image onto another. Here are a few comments:

1. The perturbation set is generally a high-dimensional polytope. Although it has a compact representation in terms of intersection of hyperplanes, it may have many more verticies, so the endeavor of attempting to characterize all the verticies of this polytope may be infeasible.

2. This technique of generating adversarial examples from combinations of image patches seems generally applicable, but it does not seems to produce good results here. The perturbations are still unnatural looking (eg. the images in Figure 7 are not exactly natural looking).

---

> ### Author Response · Authors · 2018-11-28
> **Response to your review**
>
> Thank you for your review. Below are our responses to your questions and concerns:
>
> Q1: The perturbation set is generally a high-dimensional polytope. Although it has a compact representation in terms of intersection of hyperplanes, it may have many more vertices, so the endeavor of attempting to characterize all the vertices of this polytope may be infeasible.
> A1: This is true; because the purpose of the paper is to demonstrate the existence of the polytope, our method only finds a subset of the vertices and therefore a subset of the polytope. Existence of this subset implies existence of a polytope that is as large as this subset, and so is an interesting finding. The mere existence of this subset demonstrates that an arbitrary number of examples with feature collisions can be generated.
>
> Q2: This technique of generating adversarial examples from combinations of image patches seems generally applicable, but it does not seems to produce good results here. The perturbations are still unnatural looking (eg. the images in Figure 7 are not exactly natural looking).
> A2: The purpose of our paper is *not* to generate adversarial examples, but rather to demonstrate the existence of polytopes within which all examples share similar feature activations and therefore classifications. Our goal is to find a *large* polytope that has *similar* feature activations, whereas the goal of adversarial examples is to find an image with *small* perturbation that results in a *different* classification. So, the aim is to *maximize* perturbations, not to *minimize* perturbations. The purpose of generating examples from combinations of image patches is to find corners of the polytope that are perceptually distinct, rather than to generate natural-looking images.

---

### Meta-Review · Area_Chair1 · 2018-12-14

**Confidence:** 4
**Recommendation:** Reject

**Metareview:**

The paper presents a novel view on adversarial examples, where models using
ReLU are inherently sensitive to adversarial examples because ReLU activations
yield a polytope of examples with exactly the same activation. Reviewers
found the finding interesting and novel but argue it is limited in impact.
I also found the idea interesting but the paper could probably be improved
as all reviewers have remarked. Overall, I found it borderline but probably not enough for acceptance.